# The Formation Law of Surface Profile in Fused Silica During Continuous-Wave CO_2_ Laser Processing

**DOI:** 10.3390/mi16121328

**Published:** 2025-11-26

**Authors:** Jin Zhuo, Shengfei Wang, Ting Tan, Huiliang Jin, Feng Geng, Xiangfeng Wang, Fei Fan, Qinghua Zhang, Qiao Xu

**Affiliations:** Research Center of Laser Fusion, China Academy of Engineering Physics, Mianyang 621900, China; zhuojin_113@foxmail.com (J.Z.); robertwsf@sina.com (S.W.); sattxw@163.com (T.T.); jinhl09@sina.com (H.J.); gengf0326@sina.com (F.G.); wxiangf86@163.com (X.W.); fanfei65790299@163.com (F.F.); zhangqh502@sina.com (Q.Z.)

**Keywords:** CO_2_ laser polishing, surface profile, formation law, annealing

## Abstract

Although CO_2_ laser polishing of fused silica surfaces is considered a promising advanced processing technology, its application remains fundamentally limited by low- to mid-frequency waviness and surface figure errors. To address the critical need for enhanced performance of laser-processed fused silica components, this study investigated the formation law of the surface profile during CO_2_ laser processing. Experimental results revealed the formation law governing the influence of processing parameters on surface morphology. A multivariate regression model was established to quantitatively correlate processing parameters with surface profile evolution, enabling prediction of profile errors with less than 5% deviation. Furthermore, an optimization strategy was proposed by introducing an annealing process, which reduced the surface profile error by more than 50% (from 56.94 μm to 23.18 μm). These findings provide both a theoretical basis and process guidance for the low-defect fabrication of fused silica components in high-power laser applications.

## 1. Introduction

Fused silica is highly valued for its low thermal expansion coefficient, high mechanical strength, and chemical stability. Its wide transmission window, ranging from ultraviolet to infrared wavelengths, makes it the preferred material for windows, wedge lenses, and phase elements in high-power laser systems [1]. As a typical hard and brittle material, fused silica processed by conventional grinding and polishing methods inevitably develops surface and subsurface machining defects. These defects alter the bandgap structure of the material, leading to laser-induced damage at levels significantly below its intrinsic threshold [2].

To mitigate such issues, researchers have proposed chemical etching to remove surface defects during fused silica processing [3]. However, this approach does not address the root cause of defect formation. Damage to fused silica optics at third-harmonic wavelengths remains a major challenge, thereby restricting the output capability of high-power laser facilities. Consequently, there is an urgent need for novel, low-defect manufacturing technologies to meet the requirements of high-power laser applications.

Laser polishing, a non-contact processing technique, offers advantages such as the absence of mechanical stress, reduced contamination, and wide applicability. The technology of CO_2_ laser processing of fused silica has evolved from fundamental research to a wide range of industrial and applied science fields, such as optical fabrication and surface finishing [4,5], microfluidics [6], 3D additive or subtractive manufacturing [7]. By exploiting the strong infrared absorption of fused silica, CO_2_ laser irradiation at a wavelength of 10.6 μm can rapidly fuse a thin surface layer of the material and eliminate surface defects [8]. Driven by surface tension and the Marangoni effect, the material flows from peaks to valleys, thereby rapidly reducing surface roughness and achieving nanoscale or even sub-nanoscale surface finish, thus overcoming the limitations of conventional manufacturing processes. Temple et al. at Lawrence Livermore National Laboratory [9] were the first to propose CO_2_ laser polishing for fused silica. Their results demonstrated a considerable reduction in surface defect density, with micro-defects smaller than 10 μm effectively eliminated, thereby improving the laser damage threshold. In high-power laser facilities, such as the National Ignition Facility (NIF) in the United States and the Laser MégaJoule (LMJ) in France, initial surface micro-damage on optical components can grow under subsequent laser irradiation. Localized CO_2_ laser processing can be applied to these damage sites, causing the surrounding material to re-melt and smooth out. This process effectively removes or blunts the damage sites, thereby halting their further growth and significantly extending the service life of the optical components [10]. However, Heidrich et al. [11] reported that although polishing reduced the surface roughness of ground fused silica, thermal stress introduced surface distortion. Indeed, during CO_2_ laser processing of fused silica, the material undergoes some structural modifications. The localized laser heating and subsequent cooling induce structural relaxation within the component, leading to the development of residual stress. This residual stress can subsequently modify the optical properties of the glass and adversely affect the component’s laser-induced damage threshold [12,13].

Researchers at the Hanover Laser Center in Germany explored microwave-assisted CO_2_ laser polishing [14], where samples were preheated to 700 °C using a specialized microwave heating system. This approach reduced the thermal gradient during CO_2_ laser processing. Liu Pengpeng et al. [15] used continuous-wave CO_2_ lasers to smooth fused silica surfaces and studied the influence of scanning speed on roughness. He Ting et al. [16] employed pulsed CO_2_ lasers for ablation, achieving uniform nanoscale removal by controlling the overlap rate and power density.

Although the evolution of surface defects [17], roughness reduction [18,19], and enhancement of the laser damage threshold [20] have been systematically investigated, research on the formation mechanisms and influencing factors of low-frequency errors remains limited [21]. Consequently, precise control of these key surface indicators remains challenging.

In this study, the phase error generated on the surface of optical components after laser processing is referred to as the surface profile error. With reference to the spatial frequency or spatial wavelength ranges used to specify the optical quality required of NIF optics, for a sampling length L, L > 33 mm corresponds to the low-frequency range, 33 mm > L > 0.12 mm corresponds to the mid-frequency range, and L < 0.12 mm corresponds to the high-frequency range [22]. To address this gap, the present study systematically investigated the formation law of low-frequency errors on fused silica surfaces during CO_2_ laser processing. The effects of different processing parameters on such errors were analyzed through experiments and simulations, and the underlying mechanisms were elucidated. Additionally, approaches for reducing low-frequency surface errors were proposed. These results provide crucial guidelines for enhancing the surface accuracy of laser-processed fused silica.

## 2. Materials and Methods

The experimental setup used to investigate the surface profile formation law in fused silica is shown in Figure 1. A continuous-wave (CW) CO_2_ laser was employed as the light source, with a maximum output power of 60 W and a duty cycle adjustable from 0% to 99%. After laser treatment, the fused silica components underwent a sequence of heating, holding, rapid cooling, and slow cooling between ambient temperature and the material softening point using a tubular annealing furnace.

The Gaussian output laser beam was reshaped into a near-flat-top profile using a diffractive optical element (DOE). The beam profile was measured using a beam quality analyzer. Fused silica samples (Φ 50 mm × 5 mm) were mounted on a two-dimensional motion stage, enabling computer-controlled scanning. During processing, a thermal imaging camera monitored the temperature of the irradiated area in real time to ensure it exceeded the softening point of fused silica. After processing, surface profile errors were measured using an interferometer.

## 3. Results and Discussion

### 3.1. Influence of Laser Processing Parameters on Surface Profile of Fused Silica

The evolution of the low-frequency error on the surface profile of the Φ50 mm fused silica component after CW CO_2_ laser processing is shown in Figure 2. The component’s front and rear surfaces are referred to as Surface A and Surface B, respectively. The initial peak-to-valley (PV) value of surface A was 2.76 μm, and that of surface B was 6.86 μm, as shown in Figure 2a,b, respectively. After single-sided processing of surface A, the PV value of surface A increased to 52.53 μm, with numerous folds forming at the edge, resulting in local PV values reaching several tens of microns (Figure 2c). In contrast, surface B exhibited a convex topography with a PV value of 40.57 μm and smoother edges than surface A, as shown in Figure 2d.

Different combinations of laser parameters were employed to process Surface A of the Φ50 mm fused silica components. The PV values of the surface shape of the component under different parameter combinations were recorded to evaluate the influence of laser processing parameters on the surface profile. The results are shown in Table 1. The experiments demonstrated that the PV value of the surface shape varied between 10 μm and 40 μm under the selected process parameters, which ensured complete fusion of surface defects. Furthermore, the surface shape error decreased with increasing scanning spacing and scanning speed, whereas it increased with higher laser power. Figure 3 presents the surface profiles resulting from laser processing at different scanning speeds, with the laser energy and scan spacing held constant.

A sensitivity analysis was performed based on three sets of data. Typically, sensitivity can be assessed by calculating partial derivatives or elasticity. Since discrete data points are available, and some exhibit non-monotonic behavior (e.g., PV = 31.43 at d = 0.5, slightly lower than 32.59 at d = 1), the Average Elasticity provides a more accurate reflection of the overall influence. Therefore, the Average Elasticity was calculated to evaluate the overall impact, supplemented by an analysis of Absolute Change per Unit Change for verification.

When laser energy and scan spacing were fixed, the average absolute elasticity of scanning speed with respect to PV was 0.3738, and the Absolute Change per Unit Change was 36.67. When laser duty cycle and scanning speed were fixed, the average absolute elasticity of scan spacing with respect to PV was 0.3635, and the Absolute Change per Unit Change was 6.00. When scan spacing and scanning speed were fixed, the average absolute elasticity of laser energy with respect to PV was 1.6131, and the Absolute Change per Unit Change was 74.36.

Based on both the elasticity analysis and the absolute change analysis, laser energy has the greatest influence on PV (average elasticity ≈ 1.61), followed by scanning speed (average elasticity ≈ 0.37), with scan spacing having the least influence (average elasticity ≈ 0.36). Therefore, laser energy is the most critical factor affecting PV. In both modeling and practical applications, accurate estimation and control of laser energy should be prioritized.

### 3.2. The Relationship Between Laser Processing Parameters and Surface Profile

Fused silica was processed under multiple sets of laser parameters, and surface profile data were obtained. Based on solid deformation and material flow mechanisms (including thermal expansion, stress release, and surface deformation caused by cooling contraction), these physical processes were identified as the main driving forces behind surface profile formation. To quantify their effects on the final surface profile and develop a predictive model, this study examined the relationship between laser parameters (duty cycle, scanning speed, and scanning spacing) and the resulting surface profile.

Specifically, material deformation induced by temperature gradients directly determined changes in surface shape coefficients along the XY directions. The surface profile data from laser processing were fitted with an elliptic paraboloid, expressed as:z = ax^2^ + by^2^(1)

The surface profile of the laser-processed component was fitted using Equation (1) to obtain the fitted surface. Figure 4 presents a comparison between the fitted surface and the actually measured surface profile, showing both three-dimensional surface plots and two-dimensional cross-sectional profiles. The two sets of key coefficients (i.e., a and b) characterizing the surface profile characteristics were obtained. When comparing the coefficients a and b derived from the empirical formula with those from the measured surface, the Root Mean Square Error (RMSE) values are 1.0346 × 10^−5^ and 1.6586 × 10^−5^, respectively, with corresponding R^2^ values of 0.83 and 0.95.

Analysis shows that these coefficients significantly reflect the influence of laser processing parameters. A stepwise quadratic polynomial regression analysis was performed for coefficients a and b using three laser processing parameter variables, based on the Akaike Information Criterion (AIC). Both models initially included linear terms (*p*, *v*, *d*), interaction terms (*p* × *v*, *p* × *d*, *v* × *d*), and quadratic terms (*p*^2^, *v*^2^, *d*^2^). During the stepwise regression process, variables were sequentially removed until the minimum AIC was achieved.

For the model of coefficient a, the terms *v* × *d*, *p* × *d*, *p* × *v*, *v*^2^, and *d*^2^ were removed sequentially, reducing the AIC from −243.62 to −251.56. The final retained terms included the intercept, *p, v, d,* and *p*^2^. For the model of coefficient b, the terms *v* × *d*, *p* × *d*, *p* × *v*, *d*^2^ and *p*^2^ were removed sequentially, lowering the AIC from −233.72 to −239.29. The final model retained the intercept, *p*, *v*, *d*, and *v*^2^.

In the model for coefficient a:

For laser energy *p*, the linear term coefficient was positive, and the quadratic term coefficient was negative, indicating an inverted U-shaped relationship between *p* and coefficient a—i.e., as *p* increases, coefficient a first increases and then decreases.

For scanning speed *v*, the coefficient was negative, suggesting that an increase in *v* leads to a decrease in coefficient a. For scan spacing *d*, the coefficient was negative but not statistically significant (*p*-value = 0.1562 > 0.05), indicating that *d* has no significant effect on coefficient a.

In the model for coefficient b:

Laser energy *p* had a positive coefficient and was statistically significant (*p*-value = 0.0087949 < 0.05), indicating that an increase in p leads to an increase in coefficient b. Scanning speed *v* had a negative linear coefficient and a positive quadratic coefficient, suggesting a U-shaped relationship between *v* and b—i.e., as *v* increases, b first decreases and then increases. Scan spacing *d* had a negative coefficient and was highly significant (*p*-value = 0.00024585 < 0.05), indicating that an increase in d leads to a decrease in coefficient b.

A multivariate stepwise regression analysis was used to establish a quantitative relationship model between the surface shape coefficients and key laser processing parameters.a = −2.4991 × 10^−3^ + 1.208610^−2^ × *p* − 2.0089 × 10^−4^ × *v* − 1.2886 × 10^−5^ × *d* − 1.2905 × 10^−2^ × *p*^2^(2)b = 4.6062 × 10^−4^ + 7.6170 × 10^−4^ × *p* − 1.8822 × 10^−3^ × *v* − 8.3172 × 10^−5^ × *d* + 2.1565 × 10^−3^ × *v*^2^(3)

We established a quantitative relationship between laser processing parameters (duty cycle *p*, scanning speed *v*, and scanning spacing *d*) and surface shape coefficients. These coefficients essentially represent the material deformation effects driven by thermal stress:

An increase in duty cycle (*p*), which is positively correlated with laser power, raises the temperature gradient of the melt zone and enhances material flow from the center toward the edges, thereby increasing surface deformation.

An increase in scanning speed (*v*) reduces the heat input duration, lowers the temperature gradient, and suppresses material flow, resulting in a smoother surface (the negative coefficient of v confirms this, while the *v*^2^ term accounts for nonlinear effects).

The effect of scanning spacing (*d*) involves thermal accumulation; larger spacing weakens local thermal concentration and reduces flow amplitude (the negative coefficient of d supports this).

An empirical relationship quantifying the deformation law under combined laser parameters was obtained. Validation was conducted with two parameter sets: duty cycle 45%, scanning speed 0.3 mm/s, and spacing 1 mm; and duty cycle 50%, scanning speed 0.25 mm/s, and spacing 1.5 mm. Figure 5a,b show the measured surface profiles, with PV values of 13.04 μm and 13.30 μm, respectively. Figure 5c,d show the fitted surface profiles based on the empirical relationship, with PV values of 14.31 μm and 14.49 μm, respectively. The prediction error between measured and calculated profiles was within 5%.

This correlation allows the relationship to predict the effect of material flow caused by thermocapillary forces on surface shape, providing physical guidance for process optimization.

Based on these experimental results, the formation mechanism of the surface profile during CO_2_ laser processing of fused silica was analyzed. At the initial stage of laser irradiation, the material in the heating zone underwent thermal expansion as the surface temperature rose but remained below the softening point of fused silica. The surrounding low-temperature material constrained the deformation of the high-temperature zone, limiting expansion along the x-direction and generating compressive stress. Along the z-direction, expansion occurred freely due to the unconstrained upper surface, resulting in partial uplift, as shown in Figure 6b. With continuous laser irradiation, when the central spot temperature exceeded the softening point of fused silica, the viscosity decreased sharply and the compressive stress accumulated during heating was released.

Driven by thermocapillary forces, the molten material at the center of the hot-melt zone flowed outward toward the edges until the end of irradiation, as shown in Figure 6c. During subsequent cooling, contraction occurred in the high-temperature zone, with the degree of shrinkage proportional to the temperature drop. Since the melt zone had the highest temperature, it experienced the greatest shrinkage when cooled to room temperature. As a result, the upper surface deformed into a concave shape (Figure 6d), while the bottom surface remained relatively unchanged, producing the observed concave upper and convex lower morphology, as shown in Figure 2c,d.

### 3.3. Research on the Suppression Approach of Surface Profile Deformation

Laser processing serves as a non-contact approach to fuse surface and subsurface defects on optical components, enhancing their damage threshold. However, the process would introduce low-frequency surface errors. Annealing is subsequently employed to mitigate this effect. This section investigates approaches for suppressing the surface deformation induced by laser processing.

The fused silica component has completed laser processing on surface A first and then surface B. After further laser processing of surface B, the PV value of surface A decreased to 37.10 μm while maintaining its concave topography, as shown in Figure 7a. However, the PV value of surface B decreased only slightly to 40.86 μm, with numerous folds appearing at its edge, as shown in Figure 7b.

T. Doualle et al. [23] demonstrated that annealing improves the damage threshold and reduces stress in CO_2_-laser-processed fused silica. Based on the analysis of the formation law of surface profile on laser-processed fused silica conducted in this research, an annealing process was developed for the laser-processed components, as shown in Table 2. The process consisted of heating, isothermal holding, slow cooling, and fast cooling. The annealing process commenced by heating the fused silica to the target temperature at a controlled rate of 43.2 °C/h. After a 10 h hold at this temperature, the slow cooling stage was initiated, cooling the specimen to 750 °C at a rate of 15 °C/h. This was followed by the fast cooling first stage, where the cooling rate was increased to 35 °C/h until the temperature reached 350 °C. Considering efficiency requirements, upon reaching 350 °C, the fast cooling second stage was implemented, utilizing the furnace’s maximum achievable cooling rate of 300 °C/h to cool the specimen to room temperature. By heating to an appropriate temperature and holding for sufficient time, internal stresses were relieved and surface deformation corrected, thereby reducing low-frequency surface errors.

Figure 8 illustrates the deformation of components after annealing. At 950 °C with 10 h preservation, the PV values slightly decreased from 95.21 μm to 91.73 μm, showing minimal improvement. At 1080 °C with the same holding time, the PV values decreased from 56.94 μm to 23.18 μm, indicating a significant reduction of over 50%. At 1150 °C with 10 h preservation, numerous bubbles formed inside the material, leading to opacity. The results demonstrate that appropriate annealing can effectively reduce low-frequency errors, while insufficient temperature yields limited improvement, and excessive temperature induces bubble formation and opacity.

## 4. Conclusions

In this study, the formation law of laser-processed fused silica surface profiles was investigated experimentally. The main conclusions are as follows:(1)The upper surface of fused silica processed by a CW CO_2_ laser exhibited a concave morphology, whereas the bottom surface showed a convex shape. The PV value ranged between 10 and 30 μm when surface defects were fully fused. The PV value decreased with increasing scanning spacing and scanning speed but increased with higher laser power. Stress relief and cooling contraction of the hot-melt zone were the main causes of these surface profiles.(2)An empirical relationship for surface shape coefficients was proposed, providing an effective method to precisely control the surface profile of laser-processed fused silica.(3)Under annealing conditions of 1080 °C for 10 h, the low-frequency surface profile error of laser-processed fused silica was reduced by more than 50%.

This study clarified the formation law of surface profiles through experiments and analysis, laying a theoretical and technical foundation for the low-defect fabrication of fused silica components in high-power laser systems. The findings further contribute to enhancing the performance and reliability of laser-processed fused silica optics to meet the stringent requirements of high-power laser applications.

## Figures and Tables

**Figure 1 micromachines-16-01328-f001:**
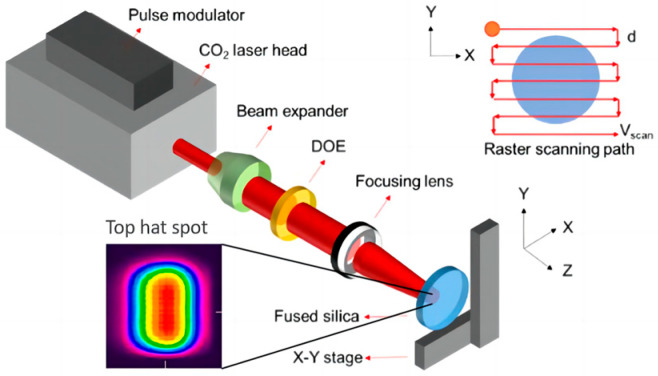
CO_2_ laser processing experimental setup.

**Figure 2 micromachines-16-01328-f002:**
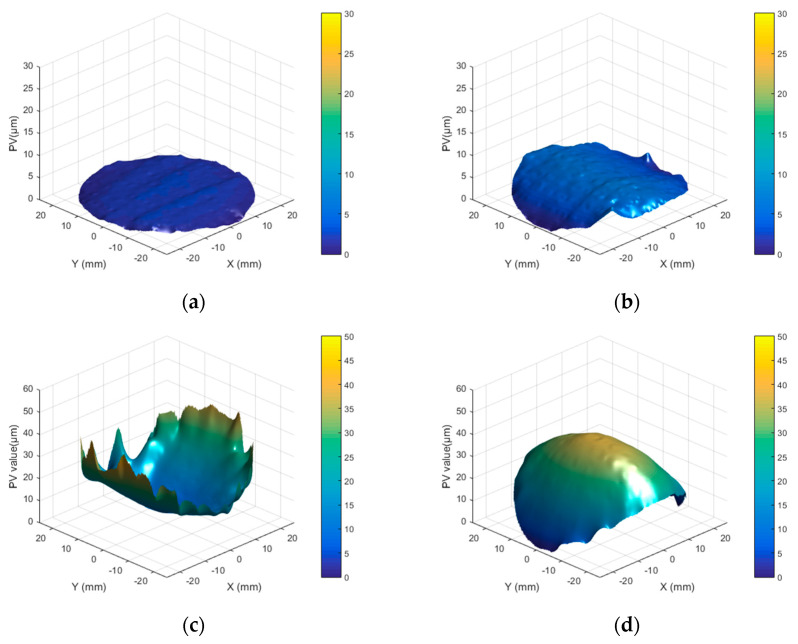
Transformation of the surface profile of the Φ50 mm fused silica component: (**a**) Initial PV value of surface A; (**b**) initial PV value of surface B; (**c**) PV value of surface A after laser processing surface A; (**d**) PV value of surface B after laser processing surface A.

**Figure 3 micromachines-16-01328-f003:**
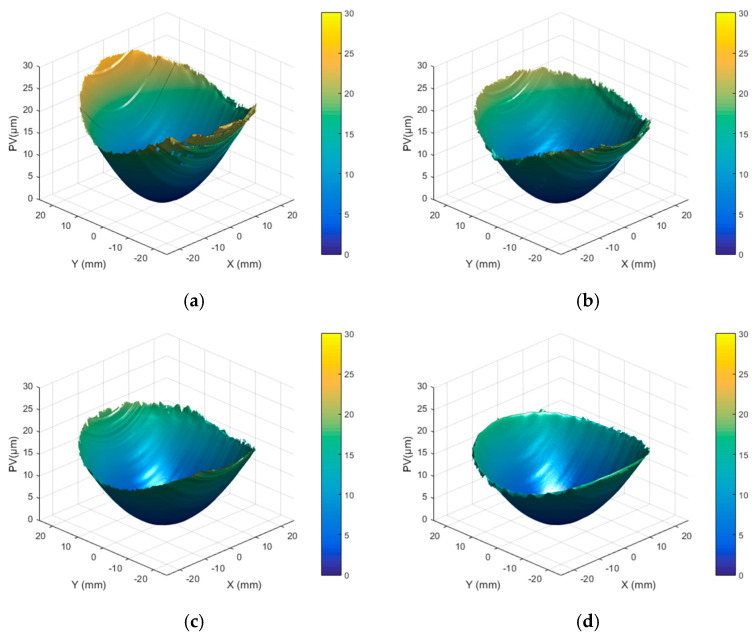
Surface A profile of fused silica under different scanning speeds: (**a**) 0.10 mm/s; (**b**) 0.20 mm/s; (**c**) 0.25 mm/s; (**d**) 0.30 mm/s.

**Figure 4 micromachines-16-01328-f004:**
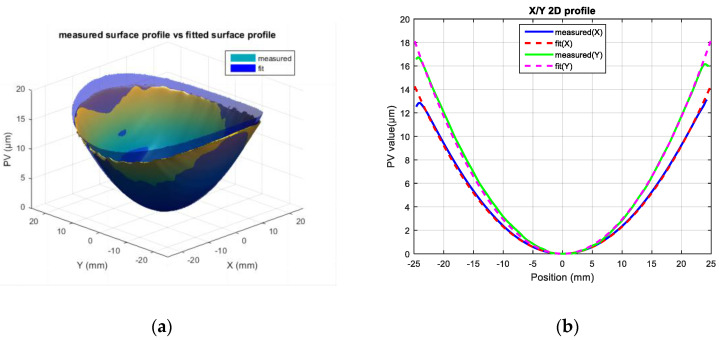
The surface profile of the fused silica component, which was an actual measurement and fitting: (**a**) comparison between measured surface profile and fitted surface profile; (**b**) comparison between measured 2D profile and fitted 2D profile.

**Figure 5 micromachines-16-01328-f005:**
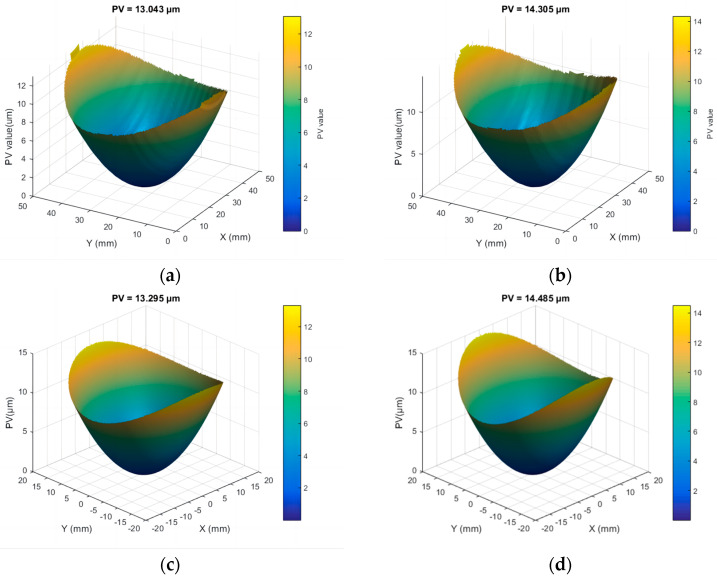
The surface profile of the fused silica component, which was an actual measurement and fitting: (**a**) measured PV with processing parameters: *p* = 45%, *v* = 0.3 mm/s, *d* = 0.25 mm; (**b**) measured PV with processing parameters: *p* = 50%, *v* = 0.25 mm/s, *d* = 1.5 mm; (**c**) fitted PV with processing parameters: *p* = 45%, *v* = 0.3 mm/s, *d* = 0.25 mm; (**d**) fitted PV with processing parameters: *p* = 50%, *v* = 0.25 mm/s, *d* = 1.5 mm.

**Figure 6 micromachines-16-01328-f006:**
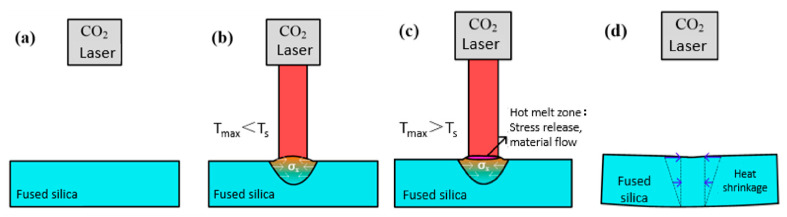
Formation process of the low-frequency profile of fused silica during laser processing: (**a**) the surface profile before CO_2_ laser processing of fused silica; (**b**) the material in the heating zone underwent thermal expansion as temperature remained below the softening point; (**c**) the heating zone temperature exceeded the softening point; (**d**) contraction occurred in the high-temperature zone during subsequent cooling.

**Figure 7 micromachines-16-01328-f007:**
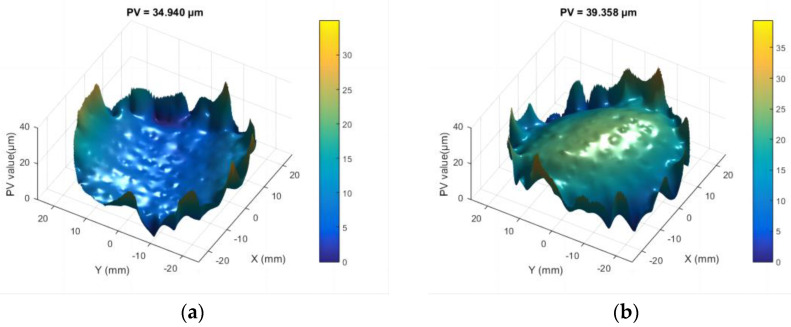
Transformation of the surface profile of the Φ50 mm fused silica component: (**a**) PV values of surface A after additional laser processing surface B; (**b**) PV values of surface B after additional laser processing of surface B.

**Figure 8 micromachines-16-01328-f008:**
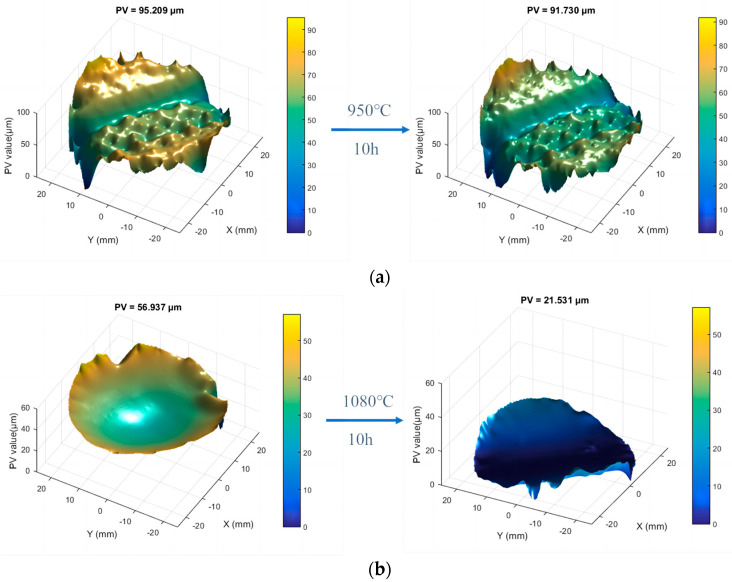
Variation in fused silica surface profile under different annealing conditions: (**a**) 950 °C, 10 h; (**b**) 1080 °C, 10 h.

**Table 1 micromachines-16-01328-t001:** Surface profile of fused silica under different laser processing parameters.

	Duty Cycle (%)	Scanning Speed (mm/s)	Scanning Spacing (mm)	PV Value (μm)
1	48	0.10	2	26.62
	48	0.20	2	22.43
	48	0.25	2	19.60
	48	0.30	2	17.74
	48	0.40	2	15.62
2	48	0.20	0.5	31.43
	48	0.20	1.0	32.59
	48	0.20	1.25	26.66
	48	0.20	1.5	24.08
	48	0.20	2	22.43
3	40	0.20	2	16.73
	42	0.20	2	17.09
	45	0.20	2	21.39
	48	0.20	2	22.43
	51	0.20	2	24.91

**Table 2 micromachines-16-01328-t002:** Experimental conditions for annealing treatment.

Heating	Heat Preservation	Slow Cooling	Fast Cooling I	Fast Cooling II
43.2 °C/h	950 °C-10 h	15 °C/h	35 °C/h	——
43.2 °C/h	1080 °C-10 h	15 °C/h	35 °C/h	——
43.2 °C/h	1150 °C-10 h	15 °C/h	35 °C/h	300 °C/h

## Data Availability

Data underlying the results presented in this paper are not publicly available at this time but may be obtained from the authors upon reasonable request.

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
