# Peer review of "The Formation Law of Surface Profile in Fused Silica During Continuous-Wave CO2 Laser Processing"

_micromachines, 2025, doi:10.3390/mi16121328_

Round 1

Reviewer 1 Report

Comments and Suggestions for Authors

This study presented a work regarding laser melting in fused silica. The author tried to find out the correlation between melted surface profile and operation parameters. Overall, the experimental results and conclusion part are reasonable. Please see my comments:

  1. At the beginning of Section 3 "This section may be divided by subheadings. It should provide a concise and precise 84 description of the experimental results, their interpretation, as well as the experimental 85 conclusions that can be drawn." I am not sure the purpose of these sentences. But I think it is not necessary.
  2. For the datapoints (duty 48%, speed 0.2mm/s and spacing 0.5), it confused me. Based on my understanding, even 0.25mm change of spacing introduced obvious change of PV value (i.g. 1.0 --> 1.25, ~ 6mm change of PV). However, nearest spacing did not generate largest PV values. And no obvious change from 0.5 --> 1.0. Did you repeat this condition to confirm? Please explain.  
  3.  Same concern with the empirical relationship you calculated. Can 48% duty, 0.2mm/s, spacing 0.5mm fit as well and generate precise prediction?
  4. Figure 6: Please clean up the text in figure. T_max 

Author Response

Dear Reviewers:

Thank you very much for taking the time to review this manuscript entitled “The Formation Law of Surface Profile in Fused Silica during Continuous-Wave CO2 Laser Processing” (ID: 3946766). Those comments are all valuable and very helpful for revising and improving our paper, as well as the important guiding significance to our researches. We have studied comments carefully and have made correction which we hope meet with approval. Please find the detailed responses below and the corresponding revisions/corrections highlighted in the re-submitted files.

Comments 1: At the beginning of Section 3 "This section may be divided by subheadings. It should provide a concise and precise 84 description of the experimental results, their interpretation, as well as the experimental 85 conclusions that can be drawn." I am not sure the purpose of these sentences. But I think it is not necessary.

Response 1: We sincerely appreciate your careful reading and thorough review. In accordance with your suggestion, we have deleted this section from the manuscript, as it pertains to content in the journal's template that is not essential to this study.

Comments 2: For the data points (duty 48%, speed 0.2mm/s and spacing 0.5), it confused me. Based on my understanding, even 0.25mm change of spacing introduced obvious change of PV value (i.g. 1.0 --> 1.25, ~ 6mm change of PV). However, nearest spacing did not generate largest PV values. And no obvious change from 0.5 --> 1.0. Did you repeat this condition to confirm? Please explain. 

Response 2: Thank you for this valuable question. It is an excellent point of inquiry. An increase in laser power or a decrease in scanning speed will raise the temperature within the processing zone. In contrast, varying the scan spacing does not alter temperature of the processing zone. In fact, a reduced scan spacing can even induce an annealing effect on areas beyond the non-molten zone due to the increased cumulative interaction time. Consequently, its influence on the resulting surface profile is limited. Furthermore, in agreement with our prior experimental data, when the laser power and scanning speed are fixed, continuously decreasing the scan spacing beyond a specific range results in no significant change—or even a slight reduction—in the surface profile.

Comments 3: Same concern with the empirical relationship you calculated. Can 48% duty, 0.2mm/s, spacing 0.5mm fit as well and generate precise prediction?

Response 3: Thank you for your attention and careful consideration. Under the laser processing condition of 48% duty, 0.2mm/s, spacing 0.5mm , the predicted PV value of the surface profile is 30.904 µm, which closely matches the measured value of 31.43 µm obtained from the actual processed component. We consider this agreement to be within an acceptable margin of error and consistent with the empirical relationship established in our study.

Comments 4: Figure 6: Please clean up the text in figure. T_max

Response 4: We feel sorry for our carelessness. In our resubmitted manuscript, the figure is revised. Thanks for your correction.

Reviewer 2 Report

Comments and Suggestions for Authors

Dear authors.

Thank you very much for interesting article.

Quality of manuscript is encouraging however some major setbacks must be solved for article to be finalized.

Those main are:

  1. From Table 1 results it is possible to give far more detail analysis of processing parameters effect on value of PV. It is possible to get very good correlation between output and inputs (at 95% and higher level). Also sensitivity analysis can be easily performed with clear result which parameter have most significant effect on PV value (at min 85% value).
  2. In chapter 3.2 you have proposed fit of PV in form of paraboloid. You also gave how constants a, b change with processing parameters. However if you do such it is must to add some statistical parameters which confirm or reject validity of such model. For most easy statistics, validity can be confirmed by R2 and RMSE value.
  3. It is not important if you use nonlinear mathematical regression models or ANN or any other regression models however more detailed analysis is needed.

Author Response

Dear Reviewers:

Thank you very much for taking the time to review this manuscript entitled “The Formation Law of Surface Profile in Fused Silica during Continuous-Wave CO2 Laser Processing” (ID: 3946766). Those comments are all valuable and very helpful for revising and improving our paper, as well as the important guiding significance to our researches. We have studied comments carefully and have made correction which we hope meet with approval. Please find the detailed responses below and the corresponding revisions/corrections highlighted in the re-submitted files.

Comments 1: [From Table 1 results it is possible to give far more detail analysis of processing parameters effect on value of PV. It is possible to get very good correlation between output and inputs (at 95% and higher level). Also sensitivity analysis can be easily performed with clear result which parameter have most significant effect on PV value (at min 85% value).]

Response 1: Thank you for your attention and careful consideration. We agree with this comment. Therefore, a sensitivity analysis was performed based on three sets of data. Typically, sensitivity can be assessed by calculating partial derivatives or elasticity. Since discrete data points are available, and some exhibit non-monotonic behavior (e.g., PV = 31.43 at d=0.5, slightly lower than 32.59 at d=1), the Average Elasticity provides a more accurate reflection of the overall influence. Therefore, the Average Elasticity was calculated to evaluate the overall impact, supplemented by an analysis of Absolute Change per Unit Change for verification. When laser energy and scan spacing were fixed, the average absolute elasticity of scanning speed with respect to PV was 0.3738, and the Absolute Change per Unit Change was 36.67.When laser duty cycle and scanning speed were fixed, the average absolute elasticity of scan spacing with respect to PV was 0.3635, and the Absolute Change per Unit Change was 6.00.When scan spacing and scanning speed were fixed, the average absolute elasticity of laser energy with respect to PV was 1.6131, and the Absolute Change per Unit Change was 74.36.

Based on both the elasticity analysis and the absolute change analysis, laser energy has the greatest influence on PV (average elasticity ≈ 1.61), followed by scanning speed (average elasticity ≈ 0.37), with scan spacing having the least influence (average elasticity ≈ 0.36). Therefore, laser energy is the most critical factor affecting PV. In both modeling and practical applications, accurate estimation and control of laser energy should be prioritized. This change can be found – page 4, paragraph 2, and line 111. 

Comments 2: [In chapter 3.2 you have proposed fit of PV in form of paraboloid. You also gave how constants a, b change with processing parameters. However if you do such it is must to add some statistical parameters which confirm or reject validity of such model. For most easy statistics, validity can be confirmed by R2 and RMSE value.]

Response 2: We sincerely thank you for this valuable comment and guidance. To provide an intuitive demonstration of the fitting accuracy achieved by the empirical formula, Figure 4(a) presents and compares three-dimensional representations of the fitted surface and the actually measured surface profile under identical parameters. Cross-sectional profiles were then obtained by slicing through the center point: the X-cross-section along the laser scanning direction and the Y-cross-section perpendicular to it. Figure 4(b) displays and compares the corresponding 2D profiles of the fitted and measured surfaces.

The fitted surface was calculated using the formula z = ax² + by². When comparing the coefficients a and b derived from the empirical formula with those from the measured surface, the Root Mean Square Error (RMSE) values are 1.0346×10⁻⁵ and 1.6586×10⁻⁵, respectively, with corresponding R² values of 0.83 and 0.95. This addition has been incorporated into the revised manuscript on page 6, paragraph 3, and line 152.

Comments 3: [It is not important if you use nonlinear mathematical regression models or ANN or any other regression models however more detailed analysis is needed.]

Response 3: We sincerely thank you for your insightful comment and careful review. In response to your point on the multi-parameter regression analysis, we have added a more detailed discussion. A stepwise quadratic polynomial regression analysis was performed for coefficients a and b using three laser processing parameter variables, based on the Akaike Information Criterion (AIC). Both models initially included linear terms (p, v, d), interaction terms (p×v, p×d, v×d), and quadratic terms (p², v², d²). During the stepwise regression process, variables were sequentially removed until the minimum AIC was achieved.

For the model of coefficient a, the terms v×d, p×d, p×v, v², and d² were removed sequentially, reducing the AIC from -243.62 to -251.56. The final retained terms included the intercept, p, v, d, and p². For the model of coefficient b, the terms v×d, p×d, p×v, d² and p² were removed sequentially, lowering the AIC from -233.72 to -239.29. The final model retained the intercept, p, v, d, and v².This expanded analysis can be found in the revised manuscript on page 6, paragraph 4, and line 160.

Reviewer 3 Report

Comments and Suggestions for Authors

This manuscript addresses an interesting topic; however, the current version lacks novelty, technical soundness, and clear scientific reasoning. Therefore, I am afraid I cannot recommend it for publication.

The following comments are provided for your reference to help improve the quality of the work for future submissions:

  1. Please clarify the relative contribution of the annealing process to the surface profile improvement compared with laser polishing. 
  2. Please clearly define the surface profile error used in this study.
  3. What is the reference surface for calculating this error- is it the designed surface profile or another baseline?
  4. Please specify the designed cooling rates during the rapid and slow cooling stages.
  5. What is the reason for selecting these particular cooling rates? Are they based on experimental constraints, prior studies, or theoretical considerations?
  6. In Section 3.1, please indicate clearly which sides correspond to A and B in the figures and analysis.
  7. It is suggested to use the same PV value range in all subfigures of Figure 2 to facilitate a direct comparison of the surface profile variation before and after laser processing.
  8. Figure 3 appears quite different from the profiles in Figure 2. Please clarify the processing conditions for both figures.
  9. The functions in Equations (2) and (3) appear inconsistent with Equation (1). Please explain how the coefficients were assigned to different parameters and ensure consistent mathematical format.
  10. What is the value for applying this method if the surface profile error after processing is higher than the initial value? 

Author Response

Dear Reviewers:

Thank you very much for taking the time to review this manuscript entitled “The Formation Law of Surface Profile in Fused Silica during Continuous-Wave CO2 Laser Processing” (ID: 3946766). Those comments are all valuable and very helpful for revising and improving our paper, as well as the important guiding significance to our researches. We have studied comments carefully and have made correction which we hope meet with approval. Please find the detailed responses below and the corresponding revisions/corrections highlighted in the re-submitted files.

Comments 1: [Please clarify the relative contribution of the annealing process to the surface profile improvement compared with laser polishing.]

Response 1: Thank you for your comment. Laser processing serves as a non-contact approach to fuse surface and subsurface defects on optical components, enhancing their damage threshold. However, the process would introduce low-frequency surface errors. Annealing is subsequently employed to mitigate this effect. This section investigates approaches for suppressing the surface deformation induced by laser processing. In the revised manuscript, a paragraph has been added on page 9, paragraph 1, and line 240 to clarify the relative contribution of the annealing process to the surface profile improvement compared with laser polishing.

Comments 2: [Please clearly define the surface profile error used in this study.]

Response 2: Thank you for your attention and thoughtful consideration. In this study, the phase error generated on the surface of optical components after laser processing is referred to as the surface profile error. With reference to the spatial frequency or spatial wavelength ranges used to specify the optical quality required of NIF optics, for a sampling length L, L > 33 mm corresponds to the low-frequency range, 33 mm > L > 0.12 mm corresponds to the mid-frequency range, and L < 0.12 mm corresponds to the high-frequency range.  This define can be found in the revised manuscript on page 2, paragraph 3, and line 60.

Comments 3: [What is the reference surface for calculating this error- is it the designed surface profile or another baseline.]

Response 3: Thank you for this valuable question. In this study, the phase error generated on the surface of optical components after laser processing is referred to as the surface profile error.

Comments 4: [Please specify the designed cooling rates during the rapid and slow cooling stages.]

Response 4: Thank you for your careful comment and reminder. The annealing process commenced by heating the fused silica to the target temperature at a controlled rate of 43.2°C/h. After a 10-hour hold at this temperature, the slow cooling stage was initiated, cooling the specimen to 750°C at a rate of 15°C/h. This was followed by the fast cooling 1 stage, where the cooling rate was increased to 35°C/h until the temperature reached 350°C. Considering efficiency requirements, upon reaching 350°C, the fast cooling 2 stage was implemented, utilizing the furnace's maximum achievable cooling rate of 300°C/h to cool the specimen to room temperature. In the revised manuscript, a paragraph has been added on page 9, paragraph 3, and line 258 to clarify the cooling stages.

Comments 5: [What is the reason for selecting these particular cooling rates? Are they based on experimental constraints, prior studies, or theoretical considerations?]

Response 5: Thank you for your question. The particular cooling rates employed in this study were determined with reference to the precise annealing procedure curve for fused silica established through experiments by Kong Min et al., as well as the technical report on annealing processes and equipment for fused silica available on the official website of Jintai High-Pressure Glass Company. The limitations of our specific equipment were also taken into consideration during this process.

Comments 6: [In Section 3.1, please indicate clearly which sides correspond to A and B in the figures and analysis.]

Response 6: Thank you for your suggestion. As noted in the revised manuscript, the component's front and rear surfaces are now referred to as Surface A and Surface B, respectively. The relevant figures and analysis have been updated in the manuscript to reflect this.

Comments 7: [It is suggested to use the same PV value range in all subfigures of Figure 2 to facilitate a direct comparison of the surface profile variation before and after laser processing.]

Response 7: Thank you for your suggestion. We have revised Figure 2 in the manuscript by adopting a uniform PV value range across the plots. This adjustment provides a more direct visual comparison of the surface profiles before and after laser processing.

Comments 8: [Figure 3 appears quite different from the profiles in Figure 2. Please clarify the processing conditions for both figures.]

Response 8: Thank you for your question. Figure 2(c) presents the surface profile of the component processed with a Gaussian laser beam, illustrating the low-frequency error introduced by the laser processing. Figure 3 displays the results obtained using a top-hat laser beam. Our initial experiments employed a Gaussian beam for fusing surface defects. However, subsequent testing revealed that processing with a top-hat laser yielded lower PV and RMS values in the mid-spatial frequency range (PSD2) compared to the Gaussian laser. Therefore, this study subsequently focuses on investigating the influence and variation trends of low-frequency form errors induced on fused silica by a continuous-wave (CW) CO₂ laser with a top-hat beam.

Comments 9: [The functions in Equations (2) and (3) appear inconsistent with Equation (1). Please explain how the coefficients were assigned to different parameters and ensure consistent mathematical format.]

Response 9: Thank you for your comment. We will provide a clarification and supplement the manuscript accordingly. The surface profile of the laser-processed component was fitted using Equation (1), from which two sets of key coefficients (i.e., a and b) characterizing the surface profile characteristics were obtained. Analysis shows that these coefficients significantly reflect the influence of laser processing parameters. Subsequently, a stepwise quadratic polynomial regression analysis based on the Akaike Information Criterion (AIC) was performed for coefficients a and b using the three laser processing parameters as variables. This procedure generated two models initially containing linear terms (p, v, d), interaction terms (p×v, p×d, v×d), and quadratic terms (p², v², d²). During the stepwise regression process, terms were sequentially removed until the minimum AIC value was achieved, resulting in the final forms given as Equations (2) and (3). The added paragraph can be found on page 6, paragraph 4, and line 160 of the revised manuscript.

Comments 10: [What is the value for applying this method if the surface profile error after processing is higher than the initial value?]

Response10: We sincerely thank you for your insightful comment and careful review. Fused silica is a typical hard and brittle material, and when processed using traditional grinding and polishing methods, machining defects inevitably form on the surface/sub-surface of the component. The generation of processing defects will change the bandgap structure of the material, causing laser damage to components under conditions far below the intrinsic damage threshold. Laser polishing, as a non-contact processing technology, has advantages such as no mechanical stress, no pollution, and wide applicability. By utilizing the high absorption rate of fused silica in the far infrared band, CO2 laser irradiation with the wavelength of 10.6 μm can achieve rapid fusion of surface defects in fused silica components, which is expected to break through the bottleneck of existing fused silica processing and manufacturing. Currently, it is challenging to entirely avoid the low-frequency errors introduced by laser processing. Nevertheless, this study aims to explore strategies, including the optimization of laser parameter combinations and the application of annealing processes, to mitigate or even eliminate these errors.

Round 2

Reviewer 1 Report

Comments and Suggestions for Authors

All concerns have been addressed. I do not have more comments. 

Author Response

Dear Reviewer,

Thank you very much for taking the time to review our manuscript entitled “The Formation Law of Surface Profile in Fused Silica during Continuous-Wave CO₂ Laser Processing” (ID: 3946766).

In this round, we have expanded the Introduction section by adding relevant content and corresponding references. These revisions aim to provide a more comprehensive background on the development and applications of CO₂ laser processing in the field of ultra-precision manufacturing. The corresponding changes have been highlighted in blue in the resubmitted manuscript.

We sincerely appreciate your decision and the constructive comments you provided, which have greatly helped us improve the quality of our work.

Reviewer 2 Report

Comments and Suggestions for Authors

Thank you very much for improved version.

Please just add more references to minimal level between 20 to 30 references.

Otherwise i am very happy with improvements and recommend for accepting in present form after improving introduction part.

Author Response

Dear Reviewer,

Thank you very much for taking the time to review our manuscript entitled “The Formation Law of Surface Profile in Fused Silica during Continuous-Wave CO₂ Laser Processing” (ID: 3946766).

In response to your comments, we have expanded the Introduction section by adding relevant content and corresponding references. These revisions aim to provide a more comprehensive background on the development and applications of CO₂ laser processing in the field of ultra-precision manufacturing. The corresponding changes have been highlighted in blue in the resubmitted manuscript.

We sincerely appreciate your decision and the constructive comments you provided, which have greatly helped us improve the quality of our work.

Reviewer 3 Report

Comments and Suggestions for Authors

Thank you for your careful revision. 

Author Response

Dear Reviewer,

Thank you very much for taking the time to review our manuscript entitled “The Formation Law of Surface Profile in Fused Silica during Continuous-Wave CO₂ Laser Processing” (ID: 3946766).

In this round, we have enhanced the Introduction section by adding relevant content and supporting references. To improve the professionalism of the presentation, we have also carefully refined the wording throughout. These revisions aim to provide a more comprehensive background on the development and applications of CO₂ laser processing in the field of ultra-precision manufacturing. All corresponding changes have been highlighted in blue in the resubmitted files.

We sincerely appreciate your decision and the constructive comments you provided, which have greatly helped us improve the quality of our work.